# The Emerging Role of Vitamin C as a Treatment for Sepsis

**DOI:** 10.3390/nu12020292

**Published:** 2020-01-22

**Authors:** Markos G. Kashiouris, Michael L’Heureux, Casey A. Cable, Bernard J. Fisher, Stefan W. Leichtle, Alpha A. Fowler

**Affiliations:** 1Department of Internal Medicine, Division of Pulmonary Disease and Critical Care Medicine, Virginia Commonwealth University School of Medicine, 1200 E Broad St., P.O. Box 980050, Richmond, VA 23298, USA; michael.lheureux@vcuhealth.org (M.L.); casey.cable@vcuhealth.org (C.A.C.); bernard.fisher@vcuhealth.org (B.J.F.); alpha.fowler@vcuhealth.org (A.A.F.); 2Department of Surgery, Division of Acute Care Surgical Services, Virginia Commonwealth University School of Medicine, 1200 E Broad St., P.O. Box 980454, Richmond, VA 23298, USA; stefan.leichtle@vcuhealth.org

**Keywords:** vitamin C, high-dose intravenous vitamin C, HDIVC, sepsis, septic shock

## Abstract

Sepsis, a life-threatening organ dysfunction due to a dysregulated host response to infection, is a leading cause of morbidity and mortality worldwide. Decades of research have failed to identify any specific therapeutic targets outside of antibiotics, infectious source elimination, and supportive care. More recently, vitamin C has emerged as a potential therapeutic agent to treat sepsis. Vitamin C has been shown to be deficient in septic patients and the administration of high dose intravenous as opposed to oral vitamin C leads to markedly improved and elevated serum levels. Its physiologic role in sepsis includes attenuating oxidative stress and inflammation, improving vasopressor synthesis, enhancing immune cell function, improving endovascular function, and epigenetic immunologic modifications. Multiple clinical trials have demonstrated the safety of vitamin C and two recent studies have shown promising data on mortality improvement. Currently, larger randomized controlled studies are underway to validate these findings. With further study, vitamin C may become standard of care for the treatment of sepsis, but given its safety profile, current treatment can be justified with compassionate use.

## 1. Introduction

Sepsis is a medical emergency that occurs due to a dysregulated host response to infection, resulting in life-threatening organ dysfunction [1]. The incidence of sepsis continues to rise in hospitals and intensive care units worldwide: approximately 31 million cases of sepsis occur globally every year, with approximately 6 million deaths [2]. In the United States, an estimated 535 cases of sepsis occur annually per 100,000 people, accounting for more than USD 23 billion in annual U.S. hospital expenditures in 2013 [2,3]. Despite new advances in critical care support techniques, 30–45% of patients die following hospitalization with severe sepsis and septic shock [4,5]. Sepsis can affect every organ system to cause morbidity and mortality. One particularly devastating complication of sepsis is acute respiratory distress syndrome (ARDS), a severe form of sepsis-induced lung injury. Compared with other forms of acute lung injury, patients with sepsis-induced lung injury spend more days on the mechanical ventilator and have a higher mortality rate [6]. Sepsis disproportionately affects elderly patients with comorbidities, such as impaired immune function and limited functional status [7]. The most common infectious source of sepsis in patients is pneumonia, followed by intra-abdominal and endovascular infections [8].

In sepsis, reactive oxygen and nitrogen species are generated, leading to the oxidant-induced activation of transcription factors that drive potent inflammatory cytokine and chemokine expression [9]. Oxidants produced during sepsis promote endothelial dysfunction, injuring surface glycocalyx proteins (e.g., syndecan-1) as well as damaging cell membranes and promoting the expression of intercellular adhesion molecules. During sepsis, endothelial cell damage leads to the loss of microvascular barrier function, resulting in enhanced vascular permeability. Furthermore, oxidant-induced expression of inflammatory cytokines and chemokines upregulates endothelial cell surface adhesion molecules, which leads to activated neutrophil and platelet adhesion in the microvasculature. Despite uncovering the complex multicellular activation pathways that drive tissue injury, a “cure” remains elusive. Clinical trials have included over 15,000 patients and spent over one billion US dollars in randomized clinical trial costs [10]. These trials have targeted mediators of inflammation or coagulation such as statin agents [11,12], activated protein C [13], and monoclonal antibody to endotoxin [14], but have not safely reduced sepsis mortality, suggesting that single-target therapy fails to meet the challenges of treating the complex pathophysiology of sepsis. Overall, this accumulated knowledge strongly suggests that a more “pleotropic” (i.e., broad-based) form of therapy that can interrupt multiple pathways is required to uncover the “holy grail” of treating sepsis. At present, in 21st-century critical care practice, antibiotics, “source control,” and maintaining hemodynamic stability with fluid administration and vasopressors constitutes the mainstay therapy for sepsis. More recently, there has been emerging evidence for the use of vitamin C as a treatment of sepsis.

## 2. Pharmacology and Physiology of Vitamin C

L-Ascorbic acid (AA), or vitamin C (Figure 1), is a water-soluble vitamin which is found in all fruits and vegetables, but is particularly concentrated in citrus fruits, green peppers, strawberries, broccoli, green leaves, and potatoes [15]. It was first isolated between 1927 and 1930 from ox adrenals by the Nobel laureate Albert Szent-Györgyi at Cambridge University and the Mayo Clinic. Most animal species synthesize vitamin C in the kidneys or the liver. Humans, some mammals (such as gorillas, monkeys, bats, and guinea pigs), birds, and fish, however, have lost the ability to synthesize vitamin C [16]. More specifically, these species lost the ability to synthesize the l-gulono-γ-lactone oxidase (GLO) enzyme, which catalyzes the last step in the vitamin C synthesis: oxidizing l-Gulono-1-lactone into l-Ascorbic acid [16]. The human GLO gene was inactivated by random mutations somewhere between 38 and 92 million years ago [17]. This gene loss was compatible with survival, however, because of the abundance of vitamin C in the human diet [18].

### 2.1. Vitamin C Homeostasis

Since humans do not synthesize vitamin C, they depend on vitamin C dietary intake to stay alive. The human body stores approximately 1.5 g of vitamin C, and scurvy develops when the stores drop below 0.3 g, which is equivalent to a plasma concentration of less than 11 uM [19,20]. Normal plasma vitamin C concentrations range from 50 to 80 uM, but tissue concentrations can often reach millimolar levels (with the exception of saliva) [19]. Vitamin C is moved intracellularly by carrier proteins known as sodium-dependent vitamin C transporters (SVCT). Two SVCT isoforms exist—SVCT1 and SVCT 2—and these isoforms are highly conserved glycoproteins with 12 transmembrane domains [21,22]. While almost all body tissues (including white blood cells and platelets) express SVCT2, SVCT1 is expressed exclusively by intestinal epithelial cells, the proximal convoluted renal tubules, and the liver [21,22].

When taken orally, SVCT1 in the apical membrane of the small intestinal epithelial lumen (Figure 2) actively transports vitamin C into the epithelial cells [21]. Additionally, in the intestinal lumen, an oxidized form of vitamin C (dehydroascorbic acid (DHA)), is transported by the abundant facilitative glucose transporters GLUT into the intestinal epithelium [21]. More specifically, GLUT2 and GLUT8, which are expressed on the apical (luminal) membrane of the intestinal epithelium, are integral to DHA uptake [23]. Following absorption, the water-soluble vitamin C molecule distributes in the blood and the extracellular compartment. Tissues then take up the vitamin C via the SVCT2 transporters (Figure 2) [23]. Red blood cells are an exception, as they take up DHA and reduce it via the glutaredoxin protein or glutathione [24]. Finally, the kidneys freely filter vitamin C and reabsorb it via SVCT1 in the proximal convoluted renal tubules [23].

Pathologically, vitamin C deficiency presents as scurvy, but many disease states alter vitamin C homeostasis. Vitamin C concentrations are often low in acute illnesses such as myocardial infarction, acute pancreatitis, sepsis, and critical illness in general [26,27,28]. Moreover, aging humans (who have a higher baseline critical illness mortality) require more dietary vitamin C to reach a desired plasma ascorbate concentration. These observations are confirmed in animal studies that show diminished SVCT1 mRNA expression with aging, resulting in a decline in the capacity of cells to absorb vitamin C [28,29].

The pharmacokinetics of vitamin C absorption were explored in a landmark study of human volunteers [30]. The bioavailability, which is the percentage (%) of the drug that reaches the systemic circulation, is 100% for a single oral dose of 200 mg but decreases to 33% with a single dose of 1250 mg [30]. This is because the intestinal SVCT1 transporter achieves maximal saturation around 500–1000 mg (Figure 3). In addition, the bioavailability does not decrease linearly because of alterations in renal vitamin C excretion [30]. Overall, the maximum peak plasma concentration that was achieved with long-term, high-dose oral dosing (i.e., 3 g every 4 h) is 220 uM [31]. Intravenous vitamin C administration, however, can bypass the limitations of SVCT1-induced bioavailability to achieve 70-fold higher plasma concentrations compared to oral administration [31]. In contrast to oral intake, intravenous administration can achieve a peak plasma concentration up to 15,000 uM [31]. It is key to understand that most of the earlier trials did not appreciate the differential pharmacokinetics of vitamin C and made assumptions that the plasma levels achieved with oral administration are equivalent with the intravenous administration.

### 2.2. Pleiotropic Physiologic Functions of Vitamin C

Vitamin C has numerable physiologic molecular functions due to its role as an electron donor/reducing agent [25]. These include direct antioxidant properties and cofactor properties for a wide array of enzymes in a wide variety of cellular structures and organelles, resulting in varied tissue and organ-system effects (Table 1).

### 2.3. Vitamin C’s Mechanism of Action in Sepsis and ARDS

Subnormal plasma vitamin C concentrations are common in critically ill patients and, in particular, patients with sepsis [26,27,32,113]. Furthermore, lower vitamin C levels correlate with higher incidence of organ failure and worse outcomes in septic patients [93]. In fact, very low plasma vitamin C levels, averaging around 18 uM, are a predictable feature in severely septic patients [114]. This is caused by the explosive cytokine release present in sepsis, which interferes with the cellular regulation of vitamin C absorption. Seno et al. showed that inflammatory cytokines, such as TNFα and IL-1β, known to be present in sepsis, negatively regulate endothelial SVCT2 activity; this results in the depletion of intracellular vitamin C levels [115]. In addition, during this overwhelming disease state of oxidative stress and increased reactive oxygen species (ROS) production, there is increased vitamin C consumption by the somatic cells and by leukocyte turnover [40]. As Figure 2 suggests, leukocytes can have an up to 100-fold increase in the concentration of vitamin C compared to plasma. The increased production and turnover of those cells contributes to vitamin C depletion in sepsis [40].

Many of the physiologic roles of vitamin C are important in patients with sepsis. These include the key antioxidant properties of vitamin C, scavenging reactive oxygen species, repletion of other crucial body antioxidants vitamin E and glutathione [32,35,38,39,42,49,62,64,66,68,116], and cardiovascular benefits by supporting endogenous norepinephrine, dopamine, and vasopressin production [41,117,118]. Furthermore, vitamin C protects against the loss of epithelial and endothelial barriers and enhances neutrophil function in a multidimensional way. Moreover, vitamin C promotes lymphocytic and neutrophilic activity while attenuating neutrophil necrosis and NETosis (neutrophil extracellular trap), which contributes to multiorgan failure [69,70,71,72,73,74,75,76,77,78,79,80,81,82,83,119,120,121,122]. Vitamin C also regulates nuclear cellular responses to stress and hypoxia by regulating HIF-1α [40,49,52,62,123], produces NF-κB epigenetic modifications [124] through its ability to de-methylate histones [75,82,87,88,89,125,126], regulates pro-inflammatory and coagulation gene expression [28,41,42,44,123,125,126,127], and orchestrates the immune system and circulating cytokine homeostasis in pleotropic ways (Table 1). The combination of vitamin C’s vital functions and its depletion in septic states justifies the use of high-dose intravenous vitamin C (HDIVC) in the early phases of severe sepsis and septic shock [116].

Vitamin C also has effects in the septic patient that are more specific to sepsis-induced ARDS, which are summarized in Figure 4. These include enhanced lung epithelial barrier function (i.e., via claudins and occludins) and by epigenetic and transcriptional enhancement of protein-channels which regulate the alveolar fluid clearance, such as aquaporin-5, cystic fibrosis transmembrane regular (CFTR), epithelial sodium channels (ENaC) and Na^+^/ K^+^ ATPases [128,129]. There is rising evidence that HDIVC treatment in sepsis-induced ARDS results in significantly lower levels of circulating cell-free DNA [108], which have been associated with multiorgan failure. HDIVC treatment also resulted in significant reduction of plasma circulating syndecan-1, a component of the endothelial glycocalyx, whose levels closely correlate with and predict mortality in patients with severe sepsis and ARDS [107].

## 3. Clinical Trials

The initial data on the clinical use of vitamin C was obtained in animal models. Subsequently, several completed clinical trials contributed evidence for the therapeutic effects of HDIVC in human sepsis. The first study, published in 1986, treated 16 ARDS patients with intravenous vitamin C (1000 mg IV every 6 h) plus antioxidants (*N*-acetylcysteine, selenium, and vitamin E) versus 16 ARDS patients who received the standard care at that time (i.e., control group) [130]. There was a dramatic reduction in mortality in the vitamin C group—37% versus 71% in the standard care group (*p* < 0.01) [130]. A phase I trial in 2014 [114] proved that plasma vitamin C levels in patients with severe sepsis were low, almost at scorbutic levels, and that HDIVC administration had a dose-dependent effect in the prevention of multi-organ failure, as measured by the Sequential Organ Failure Assessment (SOFA) scores [131]. Patients who received a total of 200 mg/kg/day of HDIVC for 4 days (administered in 50 mg/kg/dose, every 6 h), had significantly lower SOFA scores than placebo, and even lower scores than the patients who received lower-doses of IV vitamin C (50 mg/kg/day administered at 12.5 mg/kg/dose, every 6 h for 4 days). In this trial, the patients in the HDIVC group (200 mg/kg/day) achieved plasma levels of up to 3000 uM at day 4. The patients receiving HDIVC also demonstrated statistically lower inflammatory biomarker levels (C-Reactive protein and procalcitonin) and lower thrombomodulin levels, which is a marker of endothelial injury [114].

In 2016, a retrospective before–after study of 94 patients with severe sepsis and septic shock [132] compared patients who received hydrocortisone (50 mg IV every 6 h for 7 days or until ICU discharge), thiamine (200 mg IV every 12 h for 4 days or until ICU discharge) and HDIVC (6000 mg/day, in 4 divided doses for 4 days or until ICU discharge) to control. This study showed a 31.9% decrease in absolute hospital mortality between cases who received the triple-therapy and controls (8.5% vs. 40.4% respectively). A small randomized controlled trial, performed around the same time, of 28 patients with septic shock who received moderate doses of IV vitamin C (25 mg/kg every 6 h for 3 days) showed significantly lower mortality in patients who received IV vitamin C—14.3% vs. 64.3% [117]. The same trial found a significant reduction in average norepinephrine doses, total norepinephrine doses and total duration of norepinephrine infusion [117]. A subsequent meta-analysis of the three above studies found a significant benefit of intravenous vitamin C, with “marked reduction” in mortality and duration of vasopressor administration [133].

The largest trial completed on vitamin C to date, the CITRIS-ALI trial, was published in 2019 [134]. This multicenter, randomized, double-blinded trial included 167 patients with sepsis and ARDS who were randomized to receive 50 mg/kg every 6 h of HDIVC for 4 days versus placebo and showed statistically significant difference in 28-day all-cause mortality. The 28-day mortality was 29.8% in the vitamin C group versus 46.3% in the placebo group, although this was a secondary outcome. The statistical effect on mortality remained for up to 60 days following trial completion. The most dramatic reduction in mortality was noted during the period of HDIVC infusion (Figure 5). Furthermore, the HDIVC group had a strong trend towards more ventilator-free days (13.1 in the HDIVC group vs 10.6 in the placebo group mean difference, 2.47, 95% CI −0.90–5.85, *p* = 0.15), ICU-free days to day 28 (10.7 in HDIVC group vs. 7.7, in the placebo group, *p* = 0.03), and more hospital-free days (22.6 in HDIVC group vs. 15.5, respectively, *p* = 0.04). This trial did not find significant reductions in the SOFA scores, C-reactive protein, thrombomodulin or procalcitonin. Those biomarkers and scores, however, were not measured among the patients who “graduated” early from the ICU (a group that was heavily shifted towards the HDIVC group) or in those patients who died (heavily shifted towards the placebo group), indicating a strong selection bias, which makes these results difficult to interpret. Several other randomized controlled trials of HDIVC are under way, such as the VICTAS trial, and the Clinical Trials Network for the Prevention and Early Treatment of Acute Lung Injury (PETAL Network) is currently planning a randomized controlled trial of HDIVC for the prevention of ARDS.

## 4. Adverse Effects of Vitamin C Therapy

In all the sepsis trials mentioned above, HDIVC was found to be safe and no significant side-effects were identified. Additionally, two studies in non-medical patients did not report adverse side effects. The first was a study of infused vitamin C, 1000 mg every 8 h, combined with oral vitamin E for 28 days in 594 surgically critically ill patients and found a significantly lower incidence of acute lung injury and multiorgan failure, with no side effects [35]. The second was a study of infused vitamin C continuously at 66 mg/kg per hour for the first 24 h in patients with greater than 50% surface area burns showed that the therapy was well-tolerated with no reported side effects [135]. One proposed side effect of HDIVC is an increased propensity for oxalate kidney stone production, but this has not been shown in any clinical trials to date.

One consideration in utilizing vitamin C is that it is thought to cause an artefactual rise in point-of-care blood glucose readings by nearly all point-of-care devices [136,137]. It does not, however, raise blood glucose readings from a basic metabolic panel or glucose results using blood gas laboratories that employ hexokinase technology for analysis. This finding has recently been questioned by a review of five patients where the artifact was not appreciated [138]. For now, care must be taken to assure an accurate blood glucose level from a metabolic laboratory (i.e., basic metabolic panel (BMP)) or arterial blood gas panel (ABG) before initiating any insulin therapy, given the risk of hypoglycemia due to incorrect dosage of insulin from artefactual glucometer readings.

## 5. Conclusions

A plethora of laboratory, animal, and clinical studies are building a compelling case for a crucial role of HDIVC in the treatment of sepsis. Given the multitude of mechanisms of action, vitamin C may succeed where other possible sepsis treatments have previously failed, or facilitate the success of a multi-modal approach. Not all vitamin C treatments, however, are created equal. Because of limitations in bioavailability, oral administration does not allow for the therapeutic plasma levels required in critical conditions such as sepsis, septic shock, and ARDS. Additionally, we intentionally used the acronym HDIVC to highlight that is an entirely different therapy than oral vitamin C, or low-dose intravenous administration.

Many HDIVC randomized trials are under way and at the time of publication HDIVC use in clinical practice can be used compassionately, given that it is safe, but is not yet supported by the sepsis guidelines. Clinicians should carefully appraise the existing literature, understand the pharmacokinetics, physiology and clinical evidence of HDIVC in sepsis and other syndromes, and weigh the risks and benefits of vitamin C infusion jointly with the patient and/or the patient’s family.

## Figures and Tables

**Figure 1 nutrients-12-00292-f001:**
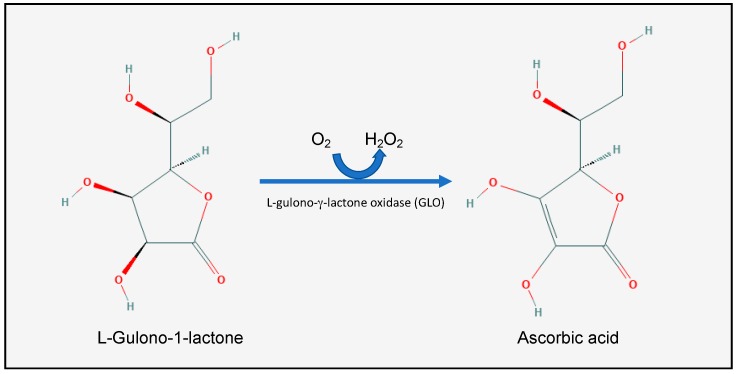
The last step in vitamin C (C_6_H_8_O_6_, or l-Ascorbic acid) biosynthesis. Humans have lost the ability to synthesize the GLO enzyme, and thus are dependent on exogenous vitamin C intake through their diet. Modified from: U.S. National Library of Medicine, PubChem.

**Figure 2 nutrients-12-00292-f002:**
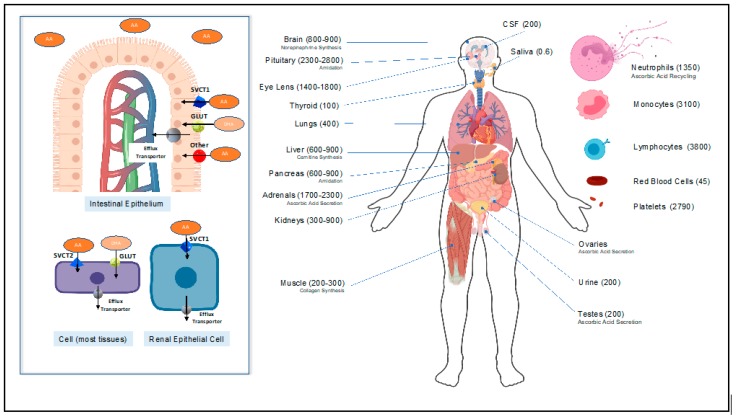
Ascorbic Acid (AA) and dehydroascorbic Acid (DHA) transporters (**left**). Concentration of AA and DHA in human organs and cells in uM (**right**). Inspired by Padayatty and Levine [25].

**Figure 3 nutrients-12-00292-f003:**
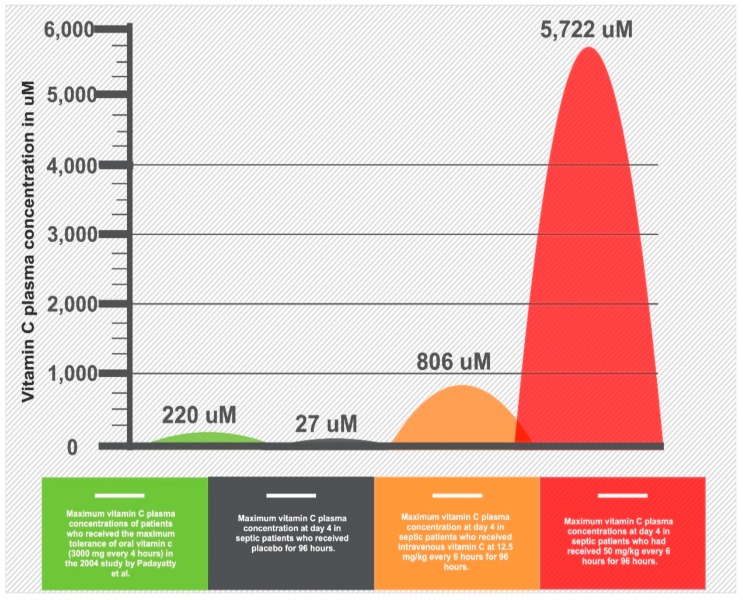
Infographic of differential vitamin C peak plasma concentrations based on alternative routes of administration and dosage.

**Figure 4 nutrients-12-00292-f004:**
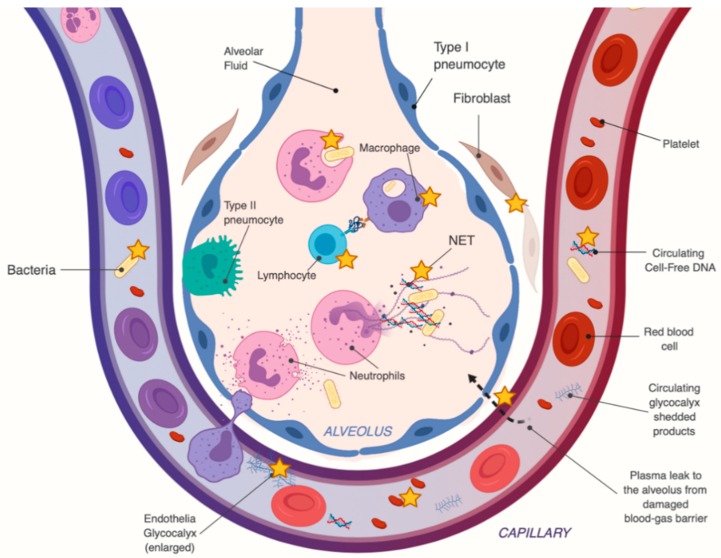
Pleiotropic effects of high-dose intravenous vitamin C (HDIVC) in sepsis-induced acute respiratory distress syndrome (ARDS). The orange star [✯] points to possible therapeutic targets of HDIVC. The figure illustrates a human alveolus with the capillary membrane, and the blood-gas barrier during sepsis.

**Figure 5 nutrients-12-00292-f005:**
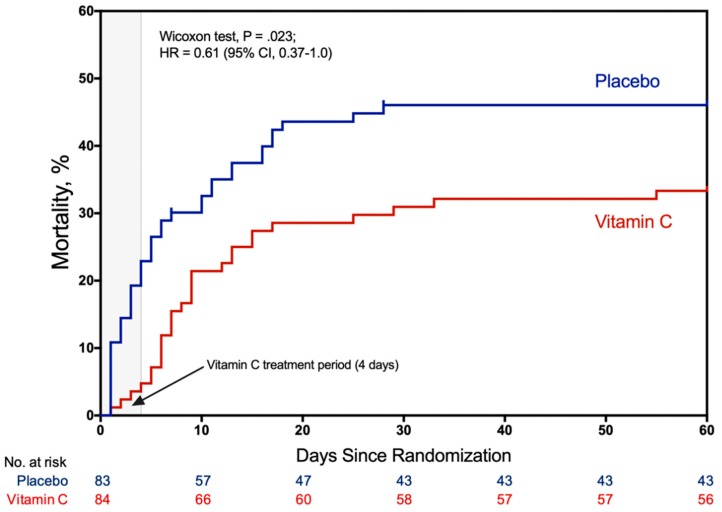
Kaplan–Meier mortality curves in patients with sepsis induced acute respiratory distress syndrome (ARDS) who were randomized to receive a 4-day course of high-dose intravenous vitamin C (HDIVC) versus placebo, upon ARDS onset-recognition.

**Table 1 nutrients-12-00292-t001:** Pleiotropic physiologic functions of vitamin C.

The key antioxidant of the body	Reverses the oxidation of lipids by the neutrophil reactive oxygen species (ROS) [32,33]. Reduces depletion of other antioxidants (e.g., vitamin E and glutathione) to prevent oxidation of lipids, proteins and DNA [34,35,36,37,38,39,40].
Norepinephrine biosynthesis	Cofactor for Dopamine ß-Hydroxylase, catalyzing the formation of norepinephrine from dopamine. Enhances adrenergic receptor activity [41].
Dopamine biosynthesis	Facilitates recycling of the enzyme cofactor tetrahydrobiopterin (BH4); a required and rate-limiting step in the hydroxylation of l-tyrosine to form l-DOPA [42,43].
Vasopressin biosynthesis	Cofactor for peptidylglycine α-amidating monooxygenase (PAM), involved in vasopressin biosynthesis [44].
Connective tissue maintenance	Vital in wound healing; cofactor for Propyl 3-hydroxylase, prolyl 4-hydroxylase, and lysyl hydroxylase which catalyze the formation of procollagen and elastin biosynthesis [45,46]. Catalyzes the hydroxylation of procollagen to form the collagen triple-helix [47]. Induces fibroblast collagen gene expression, stimulating the production of new collagen [48].
Regulation of cellular gene expression in response to hypoxia and stress	Needed for the hydroxylation (thus downregulation) of Hypoxia Induced Factor 1α (HIF-1α) by propyl and lysyl hydroxylases and FIF-1 (asparaginyl hydroxylase or factor inhibiting HIF-1) [49,50,51]. HIF-1α is a protein-transcription factor that regulates hundreds of genes in response to hypoxia and cellular stress, and is a marker of cellular hypoxia with increased expression in states of shock [52].
Carnitine biosynthesis	Cofactor for γ-butyrobetaine hydroxylase, a dioxygenase involved in carnitine synthesis, which transports fatty acids into the mitochondria [53,54]. L-Carnitine can down-modulate tumor necrosis factor (TNF-α) by endotoxins, affect lipid metabolism, and reduce septic shock severity [55].
Phagocytic cell function	Severe vitamin C deficiency (scorbutic) results in impaired neutrophilic phagocytosis and ROS generation [56,57,58,59,60]. In situations of impaired neutrophilic ROS production, vitamin C enhances the hexose monophosphate shunt (HMPS) and antibody dependent cell cytotoxicity (ADCC) resulting in increased bacterial killing [60]. Improves chemotaxis [61]. Accumulation in neutrophils may protect them from neutrophil dependent oxidative bursts [62,63]. Reduces inflammation and ROS via attenuation of NF-κB activation [64,65,66].
Inflammation: Immune cell clearance	Promotes neutrophil apoptosis, instead of necrosis via activation of caspase-3 proteins [67,68]. High-dose intravenous vitamin C (HDIVC) treatment has been shown to decrease circulating plasma cell-free DNA (resulting from neutrophil extracellular trap (NET) formations, or NETosis), and have been implicated in sepsis-induced end-organ failure [69,70,71,72,73,74,75,76,77,78,79,80,81,82,83,84].
Lymphocytic function	May promote lymphocytic proliferation, differentiation, and maturation [85,86].
Epigenetic modulation	Cofactor for ten-eleven translocation (TET) enzymes and Jumonji-C domain-containing histone demethylases (JHDMs); vitamin C increasing enzymatic activity of both, resulting in increased DNA demethylation and histone demethylation, respectively, which controls gene transcription and gene activation or repression [87,88,89].
Direct antimicrobial activity	High concentrations directly inhibit bacterial growth and exhibits bactericidal activity in vitro [90,91].
Inflammatory mediators	Modulates cytokine production and can decrease circulating histamine levels [61,92,93,94].
Endothelial function	HDIVC decreases circulating thrombomodulin, an endothelial membrane protein receptor for thrombin that converts thrombin to an anticoagulant capable of activating protein C [95]. Decreases plasma Syndecan-1 levels, a by-product of endothelial glycocalyx shedding [96,97,98,99,100,101,102,103,104,105,106,107,108].
Platelet function and Thrombosis	Alters platelet oxidative states by inhibiting CD40 ligand expression on platelet surfaces [109]. Prolonged platelet exposure to HDIVC increases Thromboxane-B2 and Prostagladin-E2 levels [110,111]. HDIVC stabilizes ADAMTS13 levels and its von-Willebrand factor cleavage activity [112].

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
