# Peer review of "The Emerging Role of Vitamin C as a Treatment for Sepsis"

_nutrients, 2020, doi:10.3390/nu12020292_

Round 1

Reviewer 1 Report

The authors describe the role of vitamin C as a treatment for sepsis in this review article. Moreover, references are too much insufficient for describing vitamin C as a treatment for sepsis as well as the conventional function of vitamin C. Even though 185 references are presented, but most of references are for Figure 4 (However, I think it is Table 1, but not Figure 4). And Figure 4 is unnecessary for this article. Reference numbers are should be revises extensively. From Reference Number 51, there are two numbers. And 50 is Bibliography, what is that? The comments are in the "Memo" of attached pdf file. 

Author Response

Reviewer 1:

The authors describe the role of vitamin C as a treatment for sepsis in this review article. Moreover, references are too much insufficient for describing vitamin C as a treatment for sepsis as well as the conventional function of vitamin C. Even though 185 references are presented, but most of references are for Figure 4 (However, I think it is Table 1, but not Figure 4). And Figure 4 is unnecessary for this article. Reference numbers are should be revises extensively. From Reference Number 51, there are two numbers. And 50 is Bibliography, what is that? The comments are in the "Memo" of attached pdf file.

Thank you for your constructive feedback, while numerous references are for Table 1, we wanted to provide detailed reference material for readers

We have removed Figure 4 given its overlap with Figure 2.

References are addressed below.  The remaining memo comments in the PDF are addressed below:

L82      References 19 and 20 have been correctly merged to [19, 20].

L82      Reference added.

L86-87 Reference added.

L90-91 Reference added.

L96-97             References added.

L100    We agree that Figure 4 is similar to the right side of Figure 2, Figure 4 has been removed.

L102    Thank you for questioning “isolated vitamin C deficiency”, we have removed ‘isolated’.

L103-104         References added.

L104-106         Reference added.

L107-108         More references added.        

L110-117         More references are needed.

Thank you for this comment, we feel that we cited the pivotal clinical studies that the literature has also historically quoted.

L123-134         Figure 3.  Quality of figure should be improved. Vitamin C concentration in Y axis is mM, but authors described as uM in figure. Y axis should be changed with uM.

            Thank you for this recommendation, we agree and will improve the quality of the figure and change the Y-axis to uM for consistency. The quality of the figure was inadvertently reduced when we attempted to embed a PDF into the Word document. This has been fixed by embedding a picture file. Please note that in addition to the PDF we include very high-resolution files (vector graphic) for all our figures as supplementary material.

L126    This paragraph in unnecessary for this review article. Moreover, I cannot understand why the authors present physiologic functions of vitamin C as a Table. Too much confuse. And so many references. I strongly recommend delete of this paragraph in this review article. Otherwise, describe with written sentences.

            We appreciate your comments.  We feel that a table presentation with bullet points offers a succinct review of what might be otherwise dense material in written sentences.  We will bring attention to the editorial office the formatting of Table 1, as there were significant formatting changes with the conversion from Word to a PDF.

L137    Figure 4 is similar as the right of Figure 2.

We agree and have removed Figure 4.

L240    The numbering of references should be revised. From reference number 51, there are two reference numbers.

            Thank you for bringing this to our attention.  These errors only appear on the formatted PFT from the editorial office.  The revision Word document provided by the editorial office does not have these errors.  It is unclear what occurred in the formatting process; however, we will bring this to their attention.

Reviewer 2 Report

It is an interesting manuscript regarding vitamin C and its role as treatment for sepsis.

The manuscript is well written and structured. The authors make a good introduction of the topic and perfectly describe the possible role of this vitamin in sepsis, as well as of the possible adverse effects derived from its therapeutic use.

Author Response

Reviewer 2:

It is an interesting manuscript regarding vitamin C and its role as a treatment for sepsis.

 The manuscript is well written and structured. The authors make a good introduction of the topic and perfectly describe the possible role of this vitamin in sepsis, as well as of the possible adverse effects derived from its therapeutic use.

Thank you for your feedback. Please find attached the latest version of the manuscript. 

Reviewer 3 Report

In the review manuscript “The emerging role of vitamin C as a treatment for sepsis” by Kashiouris et al., the authors conduct a comprehensive and clear review regarding the literature on Vitamin C and its ameliorative effects on sepsis. The review provides a good balance between basic Vitamin C biology as well as concrete examples of clinical findings. The review should garner a wide audience from both the basic and clinical researchers in the field of sepsis. This reviewer found the manuscript easily to read and the references provided within very helpful. I have only two minor comments regarding typos found in the manuscript.

Line 74. Figure 1. In the figure, “L-gulono-lactone oxidase” the Greek letter “gamma” is replaced with a box. Please correct this typo.

Line 82. The reference “[19][20]” should be changed to “[19,20]”

Author Response

Reviewer 3:

In the review manuscript “The emerging role of vitamin C as a treatment for sepsis” by Kashiouris et al., the authors conduct a comprehensive and clear review regarding the literature on Vitamin C and its ameliorative effects on sepsis. The review provides a good balance between basic Vitamin C biology as well as concrete examples of clinical findings. The review should garner a wide audience from both the basic and clinical researchers in the field of sepsis. This reviewer found the manuscript easily to read and the references provided within very helpful. I have only two minor comments regarding typos found in the manuscript.

Thank you for your comments and thoughts.

Line 74. Figure 1. In the figure, “L-gulono-lactone oxidase” the Greek letter “gamma” is replaced with a box. Please correct this typo.

            Thank you for pointing this out.  The Greek letter appears correctly in the revision Word document from the editorial office as well as the individual PDF of Figure 1 we submitted; however, it is replaced as a box in the PDF of the manuscript.  We are unclear why this occurred, but we will bring it to the attention of the editorial office.

Line 82. The reference “[19][20]” should be changed to “[19,20]”

            Thank you, this has been corrected.

Round 2

Reviewer 1 Report

The manuscript is editied and improved based on my suggesstion. So, it is acceptable for the pulication on Nutrient as a review article.